# A Literature Review of High-Tech Physiotherapy Interventions in the Elderly with Neurological Disorders

**DOI:** 10.3390/ijerph19159233

**Published:** 2022-07-28

**Authors:** Marios Spanakis, Ioanna Xylouri, Evridiki Patelarou, Athina Patelarou

**Affiliations:** 1Department of Nursing, School of Health Sciences, Hellenic Mediterranean University, GR-71004 Heraklion, Crete, Greece; iwannaxyl@hotmail.gr (I.X.); epatelarou@hmu.gr (E.P.); apatelarou@hmu.gr (A.P.); 2Computational Biomedicine Laboratory, Institute of Computer Science, Foundation for Research & Technology Hellas (FORTH), GR-70013 Heraklion, Crete, Greece

**Keywords:** physiotherapy, neurological disorders, cognitive aging, stroke, Parkinson’s disease, e-health, patient empowerment

## Abstract

Neurological physiotherapy adopts a problem-based approach for each patient as determined by a thorough evaluation of the patient’s physical and mental well-being. Τhis work aims to provide a literature review of physical therapy interventions in the elderly with neurological diseases (NDs) and discuss physiotherapy procedures and methods that utilize cutting-edge technologies for which clinical studies are available. Hence, the review focuses on acute NDs (stroke), deteriorating NDs (Parkinson’s disease), and age-related cognitive impairment. The most used physiotherapy procedures on which clinical data are available are balance and gait training (robot-assisted or not), occupational therapy, classical physiotherapy, walking and treadmill training, and upper limb robot-assisted therapy. Respectively, the most often-used equipment are types of treadmills, robotic-assisted equipment (Lokomat^®^ and Gait Trainer GT1), and portable walkway systems (GAITRite^®^), along with state-of-the-art technologies of virtual reality, virtual assistants, and smartphones. The findings of this work summarize the core standard tools and procedures, but more importantly, provide a glimpse of the new era in physiotherapy with the utilization of innovative equipment tools for advanced patient monitoring and empowerment.

## 1. Introduction

The elderly population—people over 65 years of age—is growing in developed countries and physiological deterioration (also known as frailty), characterized by obvious vulnerability to adverse health outcomes, is an increasing healthcare burden [1]. This influences various areas of daily life of elderly people, with accidents such as the increase in accidental falls to be an indicator of functionality for them [2,3]. Furthermore, there is a strong correlation between age, morbidity, and disability resulting from chronic diseases, while at the same time the healthcare system tries to meet all the needs of the elderly population for optimum healthcare provision, prediction, and prevention of additional health issues, or at least improve—if not restore—their well-being [4,5]. Among chronic diseases, the neurological disorders (NDs) such as cognitive impairment, stroke, and Parkinson’s disease are among the leading causes of disability and mortality in the elderly and a huge burden not only for patients but also their family and their caregivers. Other common NDs observed in the elderly which often require specialized healthcare provision are multiple sclerosis (MS), epilepsy, myopathy, dystonia, polyneuropathy, vertigo, idiopathic tremor, myasthenic syndrome, and migraines [6]. All these conditions underline the need for qualitative and continuous healthcare services for the elderly who not only have increased healthcare needs, but also often lack of necessary financial funds to address them [4,7].

Physiotherapy deals with the support and promotion of physical recovery after physical injuries or neurological events and conditions [8]. The degeneration of the neuromuscular system over time is an inevitable part of the aging process, a condition that makes physical therapy necessary in the elderly, especially those with additional neurological disorders (NDs) [9]. In this context, physiotherapists often have to evaluate and treat elderly patients with loss of cognitive function from aging as well as balance disorders such as cerebral palsy (and movement disorders), degenerative diseases such as Parkinson’s disease, multiple sclerosis, Alzheimer’s disease, polyneuropathy (e.g., diabetic neuropathy or Guillain–Barré syndrome), peripheral nerve injuries, and acute cases such as patients recovering from stroke and, recently, elders recovering from COVID-19 [10]. Most of the therapeutic approaches for neurological rehabilitation include basic elements such as promoting normal movement, controlling abnormal muscle tone, and facilitating function [11]. Furthermore, neurological physiotherapy adopts a problem-based individualized approach, as determined from a thorough evaluation of a patient’s health status. Therefore, the treatment goals for a person recovering from a stroke may be vastly different among patients with similar NDs but different health history. Thus, the treatment approaches used depend on the individual patient and their symptoms and physiotherapy’s rehabilitation goals, so a variety of tools and standard approaches can be applied [12].

Considering all the above, it is understood that the current conditions make it necessary to treat both physical and psychological-conscious NDs [13]. Furthermore, it is essential to enhance our awareness about the effects of different rehabilitation strategies to promote multicomponent rehabilitation programs in the frail elderly and particularly in ND patients [14,15,16]. Till today, there are various approaches for physiotherapy practices for patients with chronic and progressive NDs (i.e., Parkinson’s disease, Alzheimer’s, etc.) or recovering from acute cases (i.e., stroke). Moreover, during the last 20 years, novel equipment and electronic tools have emerged as a means for advanced healthcare provision, data analysis, and/or patient empowerment for well-being management. Over this period these tools have been evaluated through clinical trials as innovative technological tools for physiotherapy sessions. The purpose of this work is to provide a literature review for a comprehensive multidisciplinary evaluation of physiotherapy practices in the elderly with acute NDs (stroke), deteriorating NDs (Parkinson’s disease), and age-related cognitive impairment. The literature review will focus mostly on cases of physiotherapy interventions with state-of-the-art equipment that has been evaluated through clinical trials available through MEDLINE and published over the last 20 years. Literature research phrases were employed such as: “Physiotherapy or physiotherapy treatment or physical therapy”; “neurological disorders or neurological patient”. Some of the keywords that were used in the literature research were: “Physiotherapy, neurological disorders, physiotherapy equipment, technology-assisted physiotherapy, personalized processes, e-health tools, Parkinson’s disease, stroke, dystonia, idiopathic tremor, dementia, age-related cognitive impairment, multiple sclerosis, myopathy” and the retrieved articles are discussed below.

## 2. Physiotherapy Interventions in the Elderly with NDs

### 2.1. Stroke

Strokes are occurring from sudden disruption of blood supply in the brain that results in disturbance of physical and mental functions. Stroke in the brain’s right side impacts the left part of body functions, along with vision and memory loss, while in the left side, it leads to right side body paralysis, along with speech disruption, slow moves, and memory loss. If the stroke occurs in the brain stem it can affect both sides and lead to the patient being in a “locked” position, unable to make any movement below the throat, as well as being unable to speak clearly.

The main rehabilitation goals using physiotherapy with kinesiotherapy programs at regular intervals in patients recovering from stroke are the kinetic utilization as soon as possible before stiffness and muscle contractions are progressed; educating the patient on how to restore or find alternatives for lost functions (i.e., speech, use of one hand to dress up, etc.), as well as constant care and support towards adoption of new motor conditions that will assist the patient to accept the current status and return to their daily routine as much as possible (Table 1 and Figure 1). Physiotherapy sessions after stroke are a continuous healthcare process and there are cases such as intensive therapy sessions for sensory and motor stimulation that are evaluated even for chronic stroke patients for improving function and use of the affected limb in their daily activities [17].

Body-weight-supported treadmill rehabilitation for balance and gait training (robot-assisted or not) is the most often-used physiotherapy technique for restoring motor function after a stroke in elders [18,19,20,21,22]. Generally, treadmill training with body-weight support seems to lead to better walking and postural results than gait training with no body-weight support and overground walking [23,24]. Regardless of the method applied, gait trainers or floor walking exercises early after stroke assist to restore gait function [23]. A typical intervention includes a 6- or 10-min walking test for evaluation of speed and endurance, respectively. Additionally, for balance, functionality, and quality of life, evaluation tests such as those of the functional independence measure, Tinetti scale, and Short Form 36 Health Survey Questionnaire (SF-36) are often employed [25,26]. Alternate approaches after stroke are trying to assess the speed and distance to gait or mobility interventions with combinations of occupational therapy that try to establish sessions that can be followed even when at home [27,28,29].

Walking on treadmills and interventions for recovery of limb function (e.g., arms, legs, ankles, etc.) with the incorporation of mirrors to improve visual-spatial ability for treating hand disabilities or other ergometric tools or stimulation techniques seem to be beneficial when added in conventional rehabilitation programs, and their usefulness is related to the level of physical and mental functionality after stroke [30,31,32,33,34,35,36,37,38,39,40]. Robotic arms or ergometers have been found to be effective tools for rehabilitation [31,35,41], while for feet, there are several approaches of foot sensors to measure the ankle’s flexion and extension motions with mixed results till today [42,43,44]. Other approaches, such as constraint-induced movement therapy (CIMT), may reflect improvements in functionality for patients with stroke, but as the authors point out, additional data are needed to better describe and relate changes with underlying mechanisms [45]. Generally, CIMT approaches were difficult to follow in terms of the patient’s compliance and the physiotherapist’s willingness to exploit them [46,47]. Repeated transcranial magnetic stimulation seems to improve frequency of motor-evoked potentials, muscle function (torque about elbow), and purposeful movement when applied early after stroke [48]. In addition, electrical stimulation, and exercise for swallow post-stroke showed promising results for patients suffering from stroke and cricopharyngeal muscle dysfunction, as well as appearing to improve functionality regarding hand grip and maybe strength of the arms or leg muscles [49,50,51,52,53,54]. In other studies, it is suggested that early modulation of perilesional oscillation coherence seems a better strategy for brain stimulation interventions [55].

Treadmills with several add-ons including virtual reality (omnidirectional treadmills) and robotic parts are often applied in elderly patients recovering from stroke as a walking rehabilitation program to improve physical function for affected body areas (e.g., shoulder and arms) and enhance as much as possible their neuroplasticity [19,56,57,58,59,60,61,62]. In this respect, till today, several randomized clinical trials tried to evaluate the walking abilities using numerous approaches and exercises on treadmills, with also the application of obstacle crossing or other cognitive distractions, to evaluate their usability as adjuvant therapies in improving walking rehabilitation among people with stroke [18,63,64,65,66,67,68,69,70,71,72]. Thus far, the results from these trials suggest that for patients with stroke, retraining gait with the use of a treadmill and a percentage of their body weight supported (e.g., Lokomat^®^ robotic assistive device) or assisted through several wearables, resulted in improved postural abilities over those with no support on their body weight [18,24,36,41,44,71,73,74,75,76]. Variations of the intensity (e.g., speed) or of the type of the exercises (e.g., obstacles-crossing training or cognitive distractions) and the virtual reality environment or incorporation of exergames seems to be also beneficial on patients’ balance, upper limb motor function, and generally in cognitive and functional outcomes in patients recovering from stroke [19,42,70,72,77,78,79,80]. Furthermore, the incorporation of treadmill and aerobic exercises as an early intervention seems to benefit patients with stroke, except for cases with large and right-sided lesions; however, additional data are needed [18,77,81,82]. Treadmills seem also to be inferior to other approaches such as dance lessons (tango, etc.) [83]. Finally, utilization of computer-based rehabilitation games (i.e., Wii Sports^TM^, Nintendo Co., Ltd., Kyoto, Japan) may provide new, flexible, and easy to use (even from home) forms of rehabilitation for improving speed and working memory for patients after stroke [84,85].

### 2.2. Parkinson’s Disease

In Parkinson’s disease, enhancing physical function and well-being, thus improving patients’ quality of life, is essential to address the dysfunctions as the disease is progressing [86,87,88]. Physiotherapy goals in Parkinson’s disease are related with maintaining, correcting, and/or improving functional ability, strength, flexibility, posture, and balance, and thus improving daily activities, breathing, and relaxation, and reducing freezing moments that may cause falls and injuries (Table 2) [89,90]. Thus far, there are several studies of non-pharmacological physiotherapy interventions among older adults usually between 60 and 80 years old with Parkinson’s disease, often accompanied with robot assistance or virtual reality tools, that have been evaluated as interventions to enhance walking and physical performance even for home sessions [91,92,93,94,95,96,97,98,99]. These procedures include a wide range of approaches, focusing on posture, upper extremity function, balance, and gait combined with the use of cognitive movement and exercise strategies to maintain and improve quality of life.

Raccagni et al. (2019) studied the impact of physiotherapy on improving motor function in patients with Parkinson’s disease, incorporating strength, flexibility, posture, and balance exercises, walking and coordination training, swing exercises, double work, and climbing stairs, which showed that patients’ gait improved significantly [100]. Follet et al. (2010) used two different brain stimulation methods to achieve improved motor function in patients with Parkinson’s disease observing the same improvement with both methods [101]. Barboza et al. (2019), studying the effects of physiotherapy and cognitive education in patients with Parkinson’s disease, concluded that both methods, either individually or in combination, have a positive effect on patients in the outcome of the disease [102]. GAITRite^®^ combined with a closed-loop visual–auditory walker that could be applied at the patient’s home appeared to be effective in improving gait and decreasing freezing, and thus enhancing functional mobility and quality of life [95]. GAITRite^®^ has also been applied to examine the impact of an obstacle’s height for elderly patients with Parkinson’s disease, revealing similar strategies in walking over tall obstacles that could be further utilized in physiotherapy sessions [97]. Robot-assisted gait training such as Gait Trainer GT1 have been demonstrated to improve factors of balance (Berg’s Balance Scale) and reaction to an unexpected shoulder pull (Nutt’s rating) [91,92,93,103]. Other effective methods of physiotherapy to improve gait that have been tested are vibration therapy [104] and spinal cord stimulation [105]. Studies on vibration therapies—despite being proposed since the 19th century—have not managed to produce sufficient evidence with possible placebo-related factors to contribute to motor function for patients with Parkinson’s disease [104]. Pallidal or subthalamic stimulation as well as spinal cord stimulation resulted in improvements in motor function for Parkinson’s disease patients [101,105]. High intensity aerobic exercise seems to help as a feasible and safe intervention in rehabilitation of those patients mainly for sleep outcomes [106,107,108]. On the other hand, interventions for rehabilitation with embedded mental practice did not show any significant difference when compared with relaxation during rehabilitation [109].

In addition to physical exercises, alternative approaches have been employed to investigate whether similar results can be achieved, such as yoga, dance (especially tango), tai chi, and qigong [110,111,112,113,114]. Kwok et al. (2019) [110] studied the effectiveness of mindfulness yoga in relation to stretching and resistance exercises in patients with NP. They concluded that yoga is just as effective as stretching and resistance exercises in improving motor dysfunction and mobility, as well as reducing anxiety and depressive symptoms that are often present in Parkinson’s disease patients. In other cases, however, although patients’ expectations were high, the results were not encouraging. Music-contingent gait training is feasible [115], while tango dance did not show any positive feedback thus far [83,116].

Regarding state-of-the-art equipment, treadmill training is often combined with a virtual reality environment to assess physical performance. The GAITRite^®^ portable walking system seems to be the equipment of choice in most studies to measure temporal and spatial parameters for gait styles [92,95,97,117]. The results seem to point out that virtual interactive environments or visual cues are beneficial for people suffering from Parkinson’s disease [94,118,119,120]. Mirelman et al. (2011) [94] employed a progressive intensive program combining a treadmill with virtual reality equipment to evaluate walking under normal or dual-task conditions or during managing physical obstacles. Cognitive function and functional performance were also assessed with participants to process multiple stimuli simultaneously and make decisions about the various obstacles on two levels while continuing to walk down the aisle. The results showed that the combination of the treadmill with the virtual reality system can be feasible in the physiotherapy of patients with Parkinson’s disease, enhancing physical performance, gait during complex challenging conditions, and aspects of cognitive function. In clinical trials from the group of Picelli et al., the robotic pacing training using the Gait Trainer GT1 further improved the gait aspects compared to physiotherapy with active joint mobilization with a moderate amount of conventional gait training, which had also positive results but at a lower level [91,92]. Other studies are also reporting similar findings when incorporating physiotherapy alone or with cognitive training [100,102,103]. These findings have important implications for understanding motor learning in the presence of disease and for treating the risk of falling of Parkinson’s disease patients and their overall quality of life (Figure 2).

### 2.3. Age-Related Cognitive Impairment

Physiotherapy in the elderly with age-related cognitive impairment is not explored as much as in cases of stroke and Parkinson’s disease (Table 3). This can be attributed to several factors including the obscureness in diagnostic criteria to separate the condition from Alzheimer’s disease or the often reluctance from elderly people to seek medical advice, considering it an expected impact of the aging process [121,122]. However, a well-known fact is that physical exercise such as aerobic fitness or other exercises enhance cognitive function and neuropsychiatric symptoms for people with cognitive decline, thus preventing functional decline of the brain, which is a natural consequence of the aging of the body (Figure 3) [123,124,125]. Interestingly, several spatial, temporal, and spatiotemporal gait parameters (i.e., gait speed) have been suggested to be a potential subclinical marker of cognitive impairment [126]. Hence, it has been shown through several studies that patients with cognitive impairment seem to benefit from interventions that incorporate physical exercise (e.g., aerobic exercises), stationary bikes with video screens, or exergames [125,127]. Especially for patients at risk, combined physical and cognitive training shows indices of a positive neuroplastic effect in mild cognitive impaired patients [127]. In cognitive frailty, physiotherapy interventions with resistance exercise training approaches are effective in improving cognitive function and physical performance in the elderly with cognitive frailty [128]. Resistance exercises (training exercises) are effective in improving mental function and physical condition in the elderly with cognitive impairment, especially episodic memory and processing speed. Furthermore, the incorporation of exercise programs that are delivered remotely through smartphones revealed promising data regarding the advantages of remote fitness applications for delivering customized exercise programs for adults aged >65 years [129]. The utilization of smartphone technologies and virtual assistant technologies (i.e., iPad, Alexa) to deliver customized multicomponent exercise programs for older adults is continuously evaluated with promising evidence from short-term perspectives or larger randomized trials [130,131,132,133,134].

In addition to the exercise interventions, the incorporation of music and dance lessons appear to be beneficial in mild cognitive cases [135,136], whereas alternative approaches such as Tao-based techniques, although very promising in theory for improving cognitive ability, ultimately did not have the expected results in improving cognition or physical mobility [117]. On the other hand, yoga as a low-risk intervention has showed some preliminary evidence of a feasible intervention that improves visuospatial functioning in patients with mild cognitive impairment [137]. Finally, there are reports that acupressure and Montessori-based activities could have a positive impact and be beneficial for these patients, especially for soothing and reducing anxiety in people with cognitive impairment [138].

### 2.4. Other Types of NDs

For other types of NDs, few clinical studies thus far focus on elderly patients and analyze if physiotherapy interventions with or without modern healthcare equipment improve symptoms and management of the disease. A randomized controlled trial for Alzheimer’s disease revealed that aerobic exercise improves physical function and cardiorespiratory fitness (i.e., VO_2_ peaks) in patients with AD [139]. Exercise may reduce decline in global cognition, mostly in cases of elderly patients with mild-to-moderate AD dementia [140]. For multiple sclerosis (MS), it is important that cognition capabilities are supported to avoid gait impairment and falls that can lead to physical injuries [141]. Home-based exercise programs support feasibility and acceptability parameters for improving cognition and mobility function in patients with MS [142]. Patients with diabetic polyneuropathy seem to benefit from strength and balance exercises regarding their functional status and confidence through short-term sessions, although quality of life was not improved overall [143]. Finally, for elders suffering from idiopathic tremor, non-invasive peripheral nerve stimulation may provide a safe, acceptable, and effective intervention for transient relief of hand tremor symptoms [144].

## 3. Discussion

The World Confederation of Physical Therapy (WCPT) emphasizes that the knowledge and experience of physiotherapists should be used to provide timely and coordinated services, including prevention, treatment/intervention, and rehabilitation, in a way that makes them accessible to older people facing, or risk experiencing, limitations in their ability to function optimally [8]. Physiotherapy is the work of an interdisciplinary team that aims to prevent functional decline and secondary disease complications and comorbidities, to compensate and adapt to residual disabilities and to maintain or restore function and daily activities. Thus, the WPCT in its policy statement mentions that physiotherapy interventions in the elderly and especially those with NDs aim to improve well-being, increase mobility, reduce potential falls, and improve patients’ daily activities [145]. Physiotherapy in the elderly with NDs is a thorough process that aims to teach, guide, and reduce the threats to any functional and cognitive variants [146,147]. Balance and gait training, with variations, are the most often applied physiotherapy interventions for rehabilitation programs after stroke or for training patients with Parkinson’s disease, whereas for age-related cognitive impairment, aerobic physical exercises are the focus of attention currently during physiotherapy sessions. State-of-the-art equipment, such as Gait Trainer GT1, Lokomat, virtual reality equipment, or walkway systems such as the GAITRite^®^ Portable Walkway system or constant current stimulators have been utilized in clinical trials mostly during rehabilitation programs for patients suffering from stroke and/or Parkinson’s disease. Occupational therapies for stroke and stretching/aerobic exercises for Parkinson’s disease as training sessions to enhance strength and balance have also been evaluated. Alternative approaches that are based on physical exercises such as dance sessions, yoga, or martial arts such as tai chi have also been evaluated for their applicability in elders with Parkinson’s, while acupressure techniques have been evaluated even in cases of cognitive impairment. These alternative approaches, although promising and well received by patients, till today have failed to provide sufficient evidence, with a partial exception of yoga sessions, regarding their effectiveness in rehabilitation programs for elders with NDs such as Parkinson’s disease or age-related cognitive impairment.

The research on “neurorehabilitation” has advanced significantly in recent decades and the number of studies focusing on the motor learning of structural and/or functional changes of the brain in the elderly are increasing [148]. The understanding of changes within the brain in cases of neuropathological conditions becomes particularly appealing when motor learning ability translates into functional ability [149]. The neuroplasticity or the ability of the brain to restructure neural connections, specifically in response to learning or experience or after injury, is a lifelong process, even among the elderly [150,151]. Regarding the involvement of physiotherapy in neurological patients, there are several treatments available for “neurorehabilitation” and an often-used intervention is neurodevelopmental treatment [152,153]. Physiotherapy for the elderly with NDs focuses on sensory–motor disorders, orthostatic control (i.e., balance) and coordination, through to motor learning and control [154]. Preventing falls, weakness, fatigue, and sarcopenia could improve elderly patients’ health and life expectancy [141,155]. In neurological patients, it has a role in immediate or acute care, when short-term intensive hospital-based physiotherapy is required to restore musculoskeletal and neurological function, limb position, and manipulation due to hypertonic or spastic muscles [156].

Several factors are related to the lack of compliance with a physical therapy regimen in the elderly or to the availability of the physiotherapy provision. These are attributed to internal and external barriers such as insufficient time, malnutrition, lack of motivation, pleasure in exercise, fear of falling, lack of social support, space for exercise, limited finances, and more. Such reasons can prevent the maximum benefits from physical therapy [157]. Cognitive impairment, such as dementia and delirium, and psychological damage, such as depression and anxiety, can also further affect the patient’s neurorehabilitation goals and outcomes [158,159]. An additional issue that affects physiotherapy practices is the impact of quarantine and other isolation measures for the COVID-19 pandemic on physical therapy in the elderly with neurological conditions. Social disconnection, especially in states with poor public health systems, often leads to reduced exercise and fewer physiotherapy sessions and puts additional burdens on patients with NDs which are comparable to traditional clinical risk factors [160]. Already there are early indications that these measures have a negative impact on various aspects of the life of the elderly, especially those with NDs [161,162,163,164]. These negative effects are enhanced for people with NDs because they need extensive care, physiotherapy, and regular activity to maintain their well-being. Here, utilization of state-of-the-art technology tools and equipment can play a key role in addressing problems of basic healthcare provision, such as neurorehabilitation. Approaches that implement remote neurorehabilitation using digitally connected interventions (e.g., minimally supervised robotic therapy) that could accompany patients with a ND during continuous care, from the hospital to their home, have already been introduced (e.g., stroke) [165].

A trend towards novel physiotherapy practices with the utilization of state-of-the-art equipment is evident nowadays, either for performing physiotherapy or for monitoring the outcomes of physiotherapy interventions (i.e., new generation robotic-assisted and/or virtual reality technology treadmills, portable walking systems, exoskeletons, etc.). The availability of randomized clinical trials that adopt these tools with or without virtual reality equipment and evaluate them, reveals that these state-of-the-art technologies are moving from being experimental towards being routine practice physiotherapy equipment with supporting evidence regarding their usability and effectiveness. Furthermore, the continuous adoption of fitness-based home video game consoles that provide rehabilitation games (e.g., Microsoft Kinect ^TM^, Nintendo Wii Sports ^TM^) as e-health tools promise a new era in physiotherapy practices and rehabilitation processes in general [84,166,167].

Healthcare provision is moving towards a new era with the utilization of novel sensors and analytic technologies. Physiotherapists, as members of the healthcare ecosystem, should be able to adopt and apply these new approaches during physiotherapy sessions and moreover be capable to train their patients to embrace them, especially in cases of elderly people with NDs for whom sometimes communication is difficult [168,169,170]. Generally, a successful application of telemedicine technologies and smart environments should meet criteria of robustness, safety, and usability for patients; to be scalable and fitting regarding social, technical, and economic aspects and infrastructures; and finally, to be able to motivate and be transparent for patients, caregivers, health professionals, and regulatory organizations in order to increase confidence in rehabilitation technology for a home model so that all stakeholders will embrace them [165,171]. Such a shift from a traditional to modern telemedicine-assisted physiotherapy methods is expected to occur and the pandemic can act as an accelerator for the adoption by patients, caregivers, and rehabilitation professionals and for market penetration of the proposed technological rehabilitation [172,173,174,175].

The current review presented available scientific data regarding physiotherapy practices in the elderly with acute (stroke), chronic (Parkinson’s disease) or age-related (cognitive impairment) NDs. It did not provide a systematic review and meta-analysis of the outcomes but only discussed available studies. NDs have a large pool of different cases and thus the quantitative parameters and endpoints for each study are different, which would complicate the generalization of the findings. A meta-analysis would need to focus on each clinical case separately, which was out of the scope of this work. Moreover, it focused on three diseases that are the most common within the literature. Considering the range of NDs and their subtypes, generalization in other cases cannot be easily extrapolated. However, as this work shows, the fact that utilization of state-of-the-art equipment is expanding, we believe that soon enough additional clinical data of the outcomes from physiotherapy interventions will be available for further analysis.

## 4. Conclusions

There is evidence that the main physiotherapy goals for the elderly recovering from stroke or managing their Parkinson’s disease can be achieved through a structural physiotherapy program that is accompanied with innovative technological equipment to promote balance or gait capabilities and enhance elders’ motor and cognitive function, thus improving their ability to perform daily routine tasks. In addition, electrical or magnetic nerve stimulation seems beneficial in cases of stroke or Parkinson’s disease. Age-related cognitive impairment seems to benefit from physiotherapy sessions with aerobic exercises. On the other hand, except for dance as an aerobic exercise for mild cognitive impairment and yoga sessions as stretching techniques for Parkinson’s disease, other alternative approaches till today have not produced any evidence regarding their usability and effectiveness in NDs, although patients often show high adherence and compliance. As the technology advances, it is expected that future studies will further establish physiotherapy protocols for each case along with utilization of continuously evolving novel equipment. Further exploitation of emerging e-health technologies that succeed in clinical testing are expected to provide novel solutions for physiotherapists and further improve the quality of life of elderly people by advancing their independence and social participation.

## Figures and Tables

**Figure 1 ijerph-19-09233-f001:**
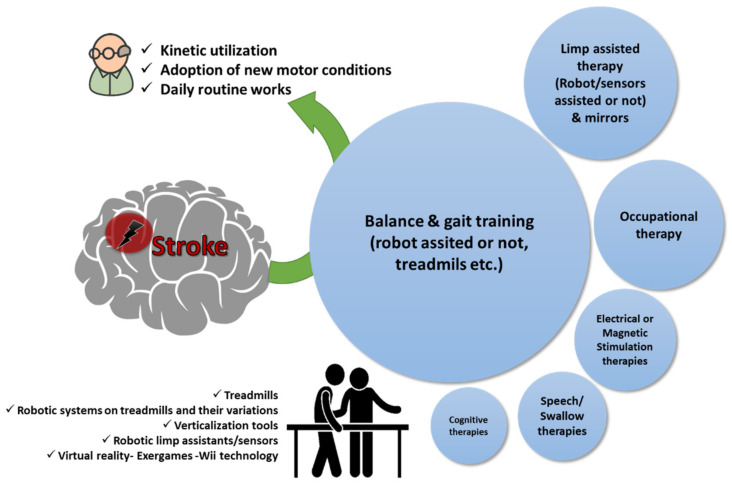
Physiotherapy goals in patients with stroke. A qualitative infographic of clinical trials in the literature regarding physiotherapy approaches in elders recovering from stroke with the incorporation of several pieces of state-of-the-art equipment.

**Figure 2 ijerph-19-09233-f002:**
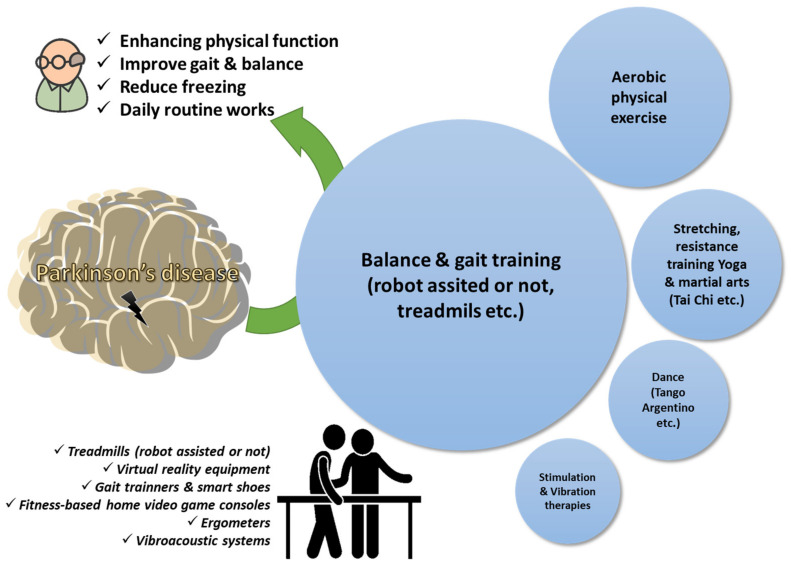
Physiotherapy goals in Parkinson’s disease. A qualitative infographic of clinical trials in the literature regarding physiotherapy approaches in elders managing Parkinson’s disease with the incorporation of several pieces of state-of-the-art equipment.

**Figure 3 ijerph-19-09233-f003:**
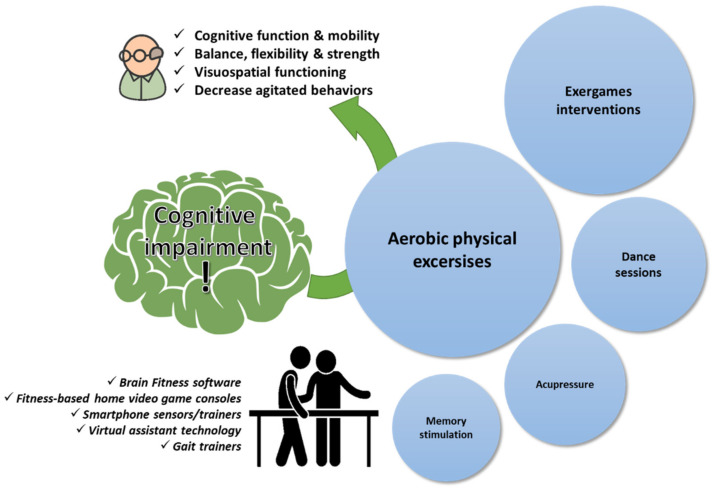
Physiotherapy goals in age-related cognitive impairment. A qualitative infographic of clinical trials in the literature regarding physiotherapy approaches in elders managing cognitive impairment with the incorporation of several pieces of state-of-the-art equipment.

**Table 1 ijerph-19-09233-t001:** Physiotherapy interventions for stroke and equipment utilized through clinical trials.

Physiotherapy Interventions	Physiotherapy’s Goal
Balance and gait training	Enhance mobility
Robot-assisted gait training	Enhance mobility
Occupational therapy and physiotherapy	Enhance mobility and limb function
Walking and treadmill training	Enhance mobility
Upper limb robot-assisted therapy	Enhance mobility and limb function
Body-weight-supported walking training	Enhance mobility
Transcranial direct current Stimulation	Enhance limb function and swallow
Electrical stimulation therapy	Enhance limb function
Arm rehabilitation therapy	Enhance mobility
Cognitive training	Enhance cognitive function
Mirror therapy	Enhance mobility
Speech and language therapy	Enhance cognitive function
Verticalization treatment	Enhance mobility and limb function
Functional electrical stimulation	Enhance mobility and limb function
Ankle treatment	Enhance mobility and limb function
Neurocognitive robot-assisted rehabilitation	Enhance mobility and limb function
Robot-assisted arm training	Enhance mobility and limb function
Swallow therapy	Enhance cognitive function and swallow
Transcranial magnetic stimulation	Enhance mobility and limb function
Continuous theta burst stimulation	Enhance cognitive function
**Equipment utilized in rehabilitation physiotherapy sessions**	
Treadmills	Wii Sports Resort
Virtual reality equipment	Stride Management Assistant
Lokomat™	Mobile hoist (ROPOX ALL IN ONE)
GAITRite^®^ Portable Walkway System	GripAble
Constant current stimulator	Com-Pressor Belt
InMotion	BIORescue
Nintendo^ΤΜ^ Wii balance board	Vibrotactile BF device
Ankle Foot Orthosis	MagPro X100 stimulator
ERIGO^®^	Wireless joint angle sensors
MIT-Manus	Stimulo
Arm ergometer	All-In-One Walking Trainer
Stationary bicycle	Nintendo^TM^ Wii Sports games
Walkbot	Electrical stimulation machines
Robot system for upper limb impairment	Robot-assisted Bi-Manu Track
Ekso wearable exoskeleton	GEAR system
Planar Robotic Manipulandum	ReHapticKnob
NeReBot	

**Table 2 ijerph-19-09233-t002:** Physiotherapy interventions in Parkinson’s disease and equipment utilized through clinical trials.

Physiotherapy Interventions	Physiotherapy’s Goal
Balance and gait training	Enhance motor function
Robot-assisted gait training	Enhance motor function
Walking and treadmill training	Enhance motor function
Aerobic dance therapy	Enhance cognitive function
Stretching and resistance training	Enhance motor and cognitive function
Body-weight-supported walking training	Enhance motor function
Physical activity and treatment therapy	Enhance motor function
Tai chi	Enhance motor function and emotional status
Argentine tango	Enhance motor function and emotional status
Obstacle-crossing training	Enhance motor function
Retraining function therapy	Enhance motor function
Spinal cord stimulation therapy	Enhance motor function
Vibration therapy	Enhance motor function
**Equipment utilized in rehabilitation physiotherapy sessions**	
Treadmills	Gamepad with wearable internal sensors
Gait Trainer GT1	Recumbent Cycle Ergometers
Virtual reality equipment	SMART Lounge vibroacoustic system
Lokomat™	Smart shoes
GAITRite^®^ Portable Walkway System	Microsoft Kinect and Xbox gaming console
XaviX system	

**Table 3 ijerph-19-09233-t003:** Physiotherapy interventions in cognitive impairment and equipment utilized through clinical trials.

Physiotherapy Interventions	Physiotherapy’s Goal
Exergames therapy	Enhance changes in the cortical activity
Aerobic dance therapy	Enhance cognitive function and mobility
Acupressure	Decrease agitated behaviors
Memory intervention	Enhance cognitive function and mobility
Exercise programs	Enhance balance, flexibility, and strength
Yoga	Enhance visuospatial functioning
**Equipment utilized in rehabilitation physiotherapy sessions**	
Brain Fitness software	GAITRite
Smartphone sensors/trainers	Stationary bikes
Amazon Alexa	iPad

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
