# Peer review of "A Literature Review of High-Tech Physiotherapy Interventions in the Elderly with Neurological Disorders"

_ijerph, 2022, doi:10.3390/ijerph19159233_

Round 1

Reviewer 1 Report

Dear Authors,

in my opinion, the topic is interesting and you provided a broad overview of the different rehabilitation strategies for specific neurological disorders, presenting a state of the art of multicomponent rehabilitation programs targeting specific physical and psychosocial impairment of patients with neurological diseases.

However, some minor revisions should be addressed in order to further improve the manuscript.

Minors

WHOLE MANUSCRIPT. A careful English revision is necessary because several typing and spacing errors should be addressed.

ABSTRACT. Page 1, line 16. Please, correct “Τhe most often physiotherapy procedures that clinical data are available are” with “Τhe most used physiotherapy procedures on which clinical data are available are”

ABSTRACT. Page 1, line 19. Please, correct “therapy. .” with “therapy.”.

PHYSIOTHERAPY INTERVENTIONS IN THE ELDERLY WITH NDS. Page 3, line 109. Please, correct “this is effect is” with “this effect is”.

PHYSIOTHERAPY INTERVENTIONS IN THE ELDERLY WITH NDS. Page 3, line 121. Please, correct “related with” with “related to”.

TABLE 1. Please, adjust the alignment of the first column.

PARKINSON’S DISEASE. Page 6, line 212. Please, correct “[110]studied” with “[110] studied”.

PARKINSON’S DISEASE. Page 6, line 218. Please, correct “[115]while” with [115] while”.

Author Response

Answer: We would like to thank the reviewer for the endorsement, the constructive comments, and the thorough reading on our work. Please find below the answers to each point that you made regarding our manuscript. They can be also tracked down in the revised version.

  1. The updated version is fully revised regarding spelling and grammar errors.
  2. Edited as suggested.
  3. Edited as suggested.
  4. Edited as suggested
  5. Table 1 edited as suggested
  6. Edited as suggested.

Reviewer 2 Report

Interesting review. Perhaps shorten the title a little.

"Physiotherapy using equipments for elderly patients with neurological conditions: A literature review"

Author Response

Answer: We would like to thank the reviewer for the endorsement of our work.  We updated the title to shorten it a little as suggested.

“A literature review of high-tech physiotherapy interventions in the elderly with neurological disorders”

We hope it suffices.

Reviewer 3 Report

Dear authors,

The Abstract does not describe methods. After the Introduction (Chapter 1), you provide the results (Chapter 3). What about Chapter 2?  You do not describe Material and Methods. It is not possible to write an article without methods. Thereover, a narrative review is not recommended for clinical trials. A systematic review is the most adequate approach (see PRISMA).

Author Response

Answer: We would like to thank the reviewer for the constructive comments, and the thorough reading on our work.

We would like to apologize for the mismatch in numbering. We edited it so now there is the correct order

  1. Introduction
  2. Physiotherapy interventions in elderly with neurological disorders
    • Stroke
    • Parkinson’s Disease
    • Age-related cognitive impairment
    • Other types of NDs
  3. Discussion
  4. Conclusion

Regarding the type of our work. We understand that a literature review provides less quantifiable evidence from a systematic review and meta-analysis when it comes to analysis of clinical trials. Our work is a is a re-submission of an initial attempt to provide a systematic review but due to substantial issues raised from the initial review evaluation we asked to provide it as a literature review. In agreement with the reviewers, we did so. We discussed available studies in the literature regarding high-tech physiotherapy interventions in the elderly with neurological disorders. If we omit any study, we would be happy to add it and discuss it in the relative section. The final paragraph in our manuscript provides additional explanation and we hope it suffices:

“The current review presented available scientific data regarding physiotherapy practices in elder with acute (stroke), chronic (Parkinson’s disease) or age related (cog-nitive impairment) NDs. It did not provide a systematic review & meta-analysis of the outcomes but only discussed available studies. NDs have a large pool of different cases thus the quantitative parameters and endpoints for each study are different which would complicate the generalization of the findings. A meta-analysis would need to focus for each clinical case separately which was out of the scope of this work. Moreover, it focused on three diseases that are the most common within the literature. Considering the range of NDs and their subtypes, generalization in other cases cannot be easily extrapolated. However, as this work shows the fact that utilization of state-of-the-art is expanding, we believe soon enough additional clinical data of the outcomes from physiotherapy interventions will be available for further analysis”.

Round 2

Reviewer 3 Report

Dear authors,

The paper does not describe Material and Methods. Regardless of the type of article, it is necessary to write a Chapter (Material and Methods) to explain step by step about the paper´s construction (For example: How do you search the articles? What are the inclusion and exclusion criteria? How many papers do you select after the search?). Therefore, a paper without a method does not show scientific evidence.

Author Response

Answer:

We would like to thank the reviewer for the comments on our work. We believe that aside from this issue there is not any other objection regarding our manuscript. We believe that this comment is within the effort to further improve our work and the reviewer in a way, agrees with the other two reviewers who recommended our work for its scientific content. In this version of our work, we follow the instructions of authors as they are available in IJERPH's page where it clearly states:

Review manuscripts should comprise the front matter (Title, Author list, Affiliations, Abstract, Keywords), literature review sections and the back matter (Supplementary Materials, Acknowledgments, Author Contributions, Conflicts of Interest, References). The template file can also be used to prepare the front and back matter of your review manuscript. It is not necessary to follow the remaining structure. Structured reviews and meta-analyses should use the same structure as research articles and ensure they conform to the PRISMA guidelines.

We understand the reviewer's point but as the journal clearly states we are not obliged to provide a detailed materials and methods section for this type of work. If we were to present an SR, then of course PRISMA guidelines should have been followed (as the reviewer proposes). But since our work is not an SR but a literature review, we believe that materials and methods only would complicate our content by creating a loop because it would shift the discussion to weather this is a suitable for publication SR or not. It is not, it is a literature review. So, in good faith and to address the issue, we further edit the last paragraph of introduction’s section: The literature review will focus mostly on cases of physiotherapy interventions with state-of-the-art equipment that have been evaluated through clinical trials available through MEDLINE and published over the last twenty years. Literature research phrases were employed such as: "Physiotherapy or physiotherapy treatment or physical therapy"; "neurological disorders or neurological patient". Some of the keywords that were used in the literature research were: “Physiotherapy, neurological disorders, physiotherapy equipment, technology-assisted physiotherapy, personalized processes, e-health tools, Parkinson’s disease, stroke, dystonia, idiopathic tremor, dementia, age related cognitive impairment, multiple sclerosis, myopathy” and the retrieved articles are discussed below.

We hope this addresses the issue raised from the reviewer.  

See also our previously published literature reviews to further support our answer.

1) Int. J. Environ. Res. Public Health 2021, 18(21), 11711; https://doi.org/10.3390/ijerph182111711

2) J. Pers. Med. 2020, 10(3), 56; https://doi.org/10.3390/jpm10030056

This manuscript is a resubmission of an earlier submission. The following is a list of the peer review reports and author responses from that submission.

Round 1

Reviewer 1 Report

Some comments are suggested:

  • It would be interesting if the keywords are DeCS/MeSH descriptors.
  • The abstract must be structured. It must also include information on Databases and the use of DeCS/MeSH descriptors.
  • In the introduction it does not make much sense to define physiotherapy, since it is a discipline already identified within the area of health sciences
  • The methodology should explain why only Pubmed and Cochrane have been used and not Metasearch engines such as EBCOhost or VHL, among others, and if google schoolar is not the most appropriate as a resource for scientific literature.
  • Access to the full text has been indicated as a criterion, what does this mean? Gratuitous or Open Access? If so, it represents a very important bias in the review.
  • The criterion of 30 years as a limit should be explained
  • The keywords were DeCS/ MeSH or free terms? This is a very important thing to set up for Boolean join algorithms.
  • In the inclusion and exclusion criteria, have you not taken into account any type of therapy or technique, procedure or characteristics of the patients included in the studies?
  • It should be explained in the methodology how the selected studies were coded?
  • No criteria have been specified to evaluate the quality of the studies based on international critical reading standards such as CASPe, CONSORT, etc.
  • In the results it is important that the studies are better contextualized: their samples, gender, average ages and deviations, countries, etc.
  • It should be clarified, if so many clinical trials and other analytical studies appear, why a meta-analysis was not carried out
  • Have the studies not been classified by their level of evidence?
  • In reality, the results of a SR are somewhat less detailed about the studies analyzed, since the discussion serves to precisely establish the similarities and differences. Even so, they can be considered like this but trying to improve the discursive coherence and the connection between paragraphs.

Reviewer 2 Report

Dear Authors,

in my opinion, the topic is interesting considering the needing to improve knowledge about the effects of different rehabilitation strategies in order to tailor multicomponent rehabilitation programs targeting specific physical and psychosocial impairment of patients with neurological diseases.

However, I have concerns about the methodological implant of the study and some critical issues should be addressed.

Major revisions

INTRODUCTION. This Section should be improved, emphasizing the needing to improve knowledge about the effects of different rehabilitation strategies in order to tailor multicomponent rehabilitation programs in frail elderly patients.

According to this, you should cite the following references:

  • Md Fadzil NH et al. A Scoping Review for Usage of Telerehabilitation among Older Adults with Mild Cognitive Impairment or Cognitive Frailty. Int J Environ Res Public Health. 2022 Mar 28;19(7):4000. doi: 10.3390/ijerph19074000.
  • de Sire A et al. Optimization of transdisciplinary management of elderly with femur proximal extremity fracture: A patient-tailored plan from orthopaedics to rehabilitation. World J Orthop. 2021 Jul 18;12(7):456-466. doi: 10.5312/wjo.v12.i7.456.
  • Zak M et al. Frailty Syndrome-Fall Risk and Rehabilitation Management Aided by Virtual Reality (VR) Technology Solutions: A Narrative Review of the Current Literature. Int J Environ Res Public Health. 2022 Mar 3;19(5):2985. doi: 10.3390/ijerph19052985.

MATERIALS AND METHODS. Please, you should use the PICO framework to highlight literature search focus.

MATERIALS AND METHODS. Please, consider updating the literature search since it has been performed a year ago.

MATERIALS AND METHODS. Please, provide a Supplementary Table with the search strategy.

MATERIALS AND METHODS. Please provide the PROSPERO registration number.

MATERIALS AND METHODS. Please, the results of articles selection process should be reported in the “Results” Section. Accordingly, Figure 1 should be reported in the “Results” section.

MATERIALS AND METHODS. Please, characterize the operators who performed the data collection/analysis.

MATERIALS AND METHODS. Please, the risk of bias assessment should be performed.

RESULTS. The results presentation should be supported by the references of the studies included.

RESULTS. Please provide a Supplementary Table including the list of full-text-assessed articles excluded with the reason of their exclusion, in order to fully meet the AMSTAR 2 quality criteria.

RESULTS. The comments to the study results should be reported in the Discussion Section rather than Results Section.

DISCUSSION. My major concern is that the research topic is too heterogeneous to be explored with a systematic review. In my opinion, a systematic review of the current relevant guidelines might improve knowledge in this field. In contrast, I have concerns about the clinical impact of a systematic review assessing different neurologic conditions (excluding other relevant ones such as spinal cord injury), a wide variability of interventions, and large variability of outcomes.

Please discuss this limitation in the limitations subsection.

Minors

INTRODUCTION. Page 2, line 62.  You start a numbered list with “i)” but do not continue it; please correct it. 

DISCUSSION. Page 10, line 314. Please, provide a definition for “WPCT”, since abbreviations used in the text should be defined at first use.

Reviewer 3 Report

Thanks for the opportunity to review the manuscript titled, " A literature review on physiotherapy interventions and utilization of equipment in the elderly with neurological disorders". The manuscript aims to explore what are the personalized physiotherapy procedures and 'stage of the art' equipment that has been tested in clinical trials for people with neurological diseases.

In general, I think the objective of the manuscript is too vague, and the manuscript lacks focus.

Some important terminologies in the objects have not been clearly defined. Which is critical for the reader to understand the scope of the review. For example, the author mentions the manuscript explores the personalized procedures in the objective. I would expect that studies selected in the study should adopt a strategy to modify the intervention to obtain a personalized treatment regime. However, I checked a few selected studies and cannot find the authors adopted any procedure to develop a personalized treatment program for the participant in the intervention group.

The authors may argue that for most of the included studies, the research personnel may modify some treatment parameters. Thus, it may be considered a 'personalized treatment.' In this case, the number of studies included in this review is surprisingly little. It is common for the rehabilitation training /protocol to be adjusted based on the patient's condition in rehab study. For example, almost all studies involving various kinds of electrical stimulation would adjust the stimulation intensity to achieve optimal muscle contraction or sensory stimulation. However, only 3 selected studies utilize electrical stimulation in this review.

The searching strategy also has a lot of shortcomings. The meanings of some search terms is vague, such as 'principles or goals'. I do not entirely understand the reasons for combining these two-term with 'standard processes or personalized processes'. Some research terms also needed to be expanded by including terms with similar meanings. For example,' personalized' is interchangeable with 'individualized.' 'e-tool' can be expanded to include 'serious game', 'telerehab' or 'computer-based program'. The current search term is not comprehensive enough to obtain all the relevant papers.

Quite a lot of arguments in the discussion session are too general, and some could be misleading, for example, in line 180-182, 'Other approaches such as constraint-induced movement therapy (CI) may reflect improvements in … needed to better describe and relate changes with underlying mechanisms' The argument regarding the mechanisms of constraints induced movement therapy was based on a study that published 20 years ago. The mechanism has been studied in this 20 years and reported. And a lot of 'additional data' have been added to the body of literature. 

In the conclusion session, the author stated, 'There is strong evidence that a …. indeed affect brain plasticity by assisting neurogenetic, neuroadaptive, and neuroprotective processes.' However, the discussion of the effect of various instruments and programs on brain plasticity is very brief. Although it is correct in general, the statement is not supported by the results and discussion of this review.

The reference number in the supplementary file should match that in the main text so the reader can quickly identify which study belongs to the selected studies.

In my opinion, the review scope of this manuscript is too broad, and the search strategy would not be able to retrieve all the relevant studies. Moreover, results and conclusions are not new to the rehabilitation research society. The manuscript provides no additional or only a minor advance in the understanding of this topic.

Reviewer 4 Report

Well presented manuscript, centered on physiotherapy interventions and utiliza
tion of equipment in the elderly with neurological disorders. Although this report could hardly be concidered a significant contribution to the field, it should be regarded as a narrative review and this should be mentioned in in the title of your manuscript. This manuscript looks like more a book chapter than a literature review that attempts to elucidate any debatable issues or compare the relative advantages and disadvantages of the aforementioned physiotherapy interventions. The main aim of such a study should not be restricted to a confirmation of the effectiveness of physiotherapy in the elderly patients with neurological diseases. Instead of that, it should compare all the available methods and propose the most effective physiotherapy protocol that is suitable for each one of the main diseases that were under concideration at your manuscript. Finally, you have mentioned that 'There is strong evidence that a structural physiotherapy program for elderly with NDs could indeed affect brain plasticity'. Ι would be very cautious with the use of the term ' brain plasticity' in this age group.